# Downregulation of IL-8 and IL-10 by the Activation of Ca^2+^-Activated K^+^ Channel K_Ca_3.1 in THP-1-Derived M_2_ Macrophages

**DOI:** 10.3390/ijms23158603

**Published:** 2022-08-03

**Authors:** Susumu Ohya, Miki Matsui, Junko Kajikuri, Hiroaki Kito, Kyoko Endo

**Affiliations:** Department of Pharmacology, Graduate School of Medical Sciences, Nagoya City University, Nagoya 467-8601, Japan; miki.matsui.4238@gmail.com (M.M.); kajikuri@med.nagoya-cu.ac.jp (J.K.); kito@med.nagoya-cu.ac.jp (H.K.); k.endo@med.nagoya-cu.ac.jp (K.E.)

**Keywords:** tumor-associated macrophage, K_Ca_3.1, K^+^ channel, THP-1, IL-8, IL-10, tumor microenvironment

## Abstract

THP-1-differentiated macrophages are useful for investigating the physiological significance of tumor-associated macrophages (TAMs). In the tumor microenvironment (TME), TAMs with the M_2_-like phenotype play a critical role in promoting cancer progression and metastasis by inhibiting the immune surveillance system. We examined the involvement of Ca^2+^-activated K^+^ channel K_Ca_3.1 in TAMs in expressing pro-tumorigenic cytokines and angiogenic growth factors. In THP-1-derived M_2_ macrophages, the expression levels of IL-8 and IL-10 were significantly decreased by treatment with the selective K_Ca_3.1 activator, SKA-121, without changes in those of VEGF and TGF-β1. Furthermore, under in vitro experimental conditions that mimic extracellular K^+^ levels in the TME, IL-8 and IL-10 levels were both significantly elevated, and these increases were reversed by combined treatment with SKA-121. Among several signaling pathways potentially involved in the transcriptional regulation of IL-8 and IL-10, respective treatments with ERK and JNK inhibitors significantly repressed their transcriptions, and treatment with SKA-121 significantly reduced the phosphorylated ERK, JNK, c-Jun, and CREB levels. These results strongly suggest that the K_Ca_3.1 activator may suppress IL-10-induced tumor immune surveillance escape and IL-8-induced tumorigenicity and metastasis by inhibiting their production from TAMs through ERK-CREB and JNK-c-Jun cascades.

## 1. Introduction

Tumor-associated macrophages (TAMs), mainly derived from monocytes, abundantly infiltrate solid tumors. They promote tumorigenesis, cancer metastasis, and the acquisition of anti-cancer drug resistance [1]. The localization of TAMs in the tumor microenvironment (TME) correlates with poor clinical outcomes in solid tumors [1]. Most TAMs have M_2_-like characteristics in the TME (up to 50% of the tumor mass in solid tumors) and secrete pro-inflammatory and anti-inflammatory cytokines/chemokines (i.e., IL-8/CXCL8 and IL-10) and angiogenic growth factors (i.e., transforming growth factor (TGF)-β1 and vascular endothelial growth factor (VEGF)) during tumor progression [2]. IL-10 promotes tumor immune surveillance escape and IL-8 is a key mediator of angiogenesis, tumorigenicity, and metastasis. Macrophages differentiated from human monocytic leukemia THP-1 cells are used as human in vitro models to study M_1_- and M_2_-polarized macrophages [3]. The manipulation of TAM functions is a useful clinical therapeutic approach to prevent tumor metastasis [4].

The intermediate-conductance Ca^2+^-activated K^+^ channel, K_Ca_3.1, decreases intracellular K^+^ concentrations ([K^+^]_i_) by facilitating K^+^ efflux, and the resulting membrane hyperpolarization increases the driving force for Ca^2+^ influx, leading to a sustained increase in intracellular Ca^2+^ concentrations ([Ca^2+^]_i_) in immune cells such as macrophages. K_Ca_3.1-mediated [Ca^2+^]_i_ increases are essential for proliferation, differentiation, apoptosis, and gene expression in immune cells and represent attractive therapeutic targets for autoimmune and inflammatory diseases [5,6]. K_Ca_3.1 indirectly regulates the production of pro- and anti-inflammatory cytokines/chemokines and macrophage polarization [7,8].

Hypoxic TME-induced death in tumor cells induces an isotonic elevation in extracellular K^+^ concentrations ([K^+^]_e_) of up to approximately 50 mM, and elevated [K^+^]_e_ within tumor interstitial fluid contributes to immunosuppression by inhibiting the functions of cytotoxic CD8^+^ T cells [9]. K^+^-driven cancer immunotherapy with a focus on K^+^ homeostasis is a potential therapeutic strategy for cancer [9,10]; however, the mechanisms underlying intracellular K^+^-driven cytokine/chemokine regulation in tumor-infiltrating immunosuppressive cells such as TAMs have not yet been elucidated.

IL-8 plays a critical role in (1) the recruitment and accumulation of TAMs and the other immunosuppressive cells, such as regulatory T (T_reg_) cells, into the TME [2,11] and (2) neovascularization/tumor angiogenesis, accelerating tumor proliferation and metastasis in the TME [12]. Increased numbers of IL-8-expressing TAMs at high levels have been detected in higher clinical stage tumors and were associated with an increased risk of a poor prognosis in cancer patients [13,14]. Bidirectional communication between M_2_ macrophages and tumor cells pathologically promotes M_2_ macrophages to secrete IL-8 into the TME. Like IL-8, the immunosuppressive cytokine IL-10 is abundantly expressed in TAMs, promoting cancer cell proliferation and metastasis [15]. Previous studies demonstrated that the transcription of IL-8 and IL-10 was critically regulated by the ERK (extracellular signal-regulated kinase) and JNK (c-Jun N-terminal kinase) signaling pathways in epithelial cells, monocytes, and fibroblasts [16,17,18]. Furthermore, several transcription factors (i.e., CREB, AP-1, and/or NF–κB) that bind to the promoters of the IL-8 and IL-10 genes have been identified [19,20].

We examined the effects of the selective K_Ca_3.1 activator, SKA-121, on the expression and secretion of IL-8 and IL-10 in THP-1-derived M_2_ macrophages. We also investigated the effects of SKA-121 on high K^+^-induced increases in IL-10/IL-8 expression and secretion in THP-1-derived M_2_ macrophages under in vitro experimental conditions mimicking the TME of solid tumors. The obtained results provide emerging mechanistic insights into the therapeutic potential of K_Ca_3.1 activators in cancer immunotherapy.

## 2. Results

### 2.1. In Vitro Differentiation of Human Monocytic THP-1 Cells into M2-like Macrophages

We initially confirmed the increased transcriptional expression of M_2_ macrophage markers in THP-1-derived M_2_-like macrophages. As reported by Javasingam et al. (2020) [21], the expression levels of CD163 and Arg1, which are widely used as markers for human M_2_ macrophages, were increased by the IL-4/IL-13-induced differentiation of M_0_ macrophages into the M_2_ phenotype (Figure 1A,B). The expression levels of the anti-inflammatory cytokine, IL-10, the pro-inflammatory chemokine, IL-8, and the angiogenic growth factor, VEGF-A, were higher in THP-1-derived M_2_ macrophages (‘M_2_’) (Figure 1C,D,E) than in native, undifferentiated THP-1 (‘native’) and PMA-differentiated M_0_ macrophages (‘M_0_’) (n = 4 for each, *p* < 0.01). The C-C motif chemokine ligands, CCL17, CCL18, and CCL22, which are secreted from activated TAMs, are associated with the enlargement of tumor size and the promotion of tumor metastasis, and the recruitment of T_reg_ cells [22]. Among these CCLs, CCL22 transcripts were highly expressed in THP-1-derived M_2_ macrophages (Figure 1F); however, the expression levels of the other CCLs were less than 0.001 in arbitrary units normalized by β-actin (ACTB) expression. TGF-β1 transcripts were highly expressed in both THP-derived ‘M_0_’ and ‘M_2_’ macrophages (Figure 1G).

### 2.2. Functional Expression of K_Ca_3.1 in THP-1-Derived M_2_ Macrophages

To perform functional analyses of K_Ca_3.1 in THP-1-derived M_2_ macrophages, we measured selective K_Ca_3.1 activator (1 μM SKA-121)-induced hyperpolarizing responses and increases in [Ca^2+^]_i_ by simultaneously monitoring DiBAC_4_(3) and Fura 2 signals, and representative time-course plots of the three different cells are shown in Figure 2A,B. In the presence of SKA-121, the selective K_Ca_3.1 blocker, TRAM-34 (1 µM), almost completely suppressed them (Figure 2A,B). SKA-121-induced hyperpolarization responses positively correlated with increase in [Ca^2+^]_i_ (correlation coefficient, R = 0.81, n = 14). The application of TRAM-34 alone did not elicit any significant changes in the membrane potential in THP-1-derived M_2_ macrophages, compared with the vehicle control (Appendix A). Additionally, SKA-121-induced [Ca^2+^]_i_ increases almost disappeared with the removal of extracellular Ca^2+^ (Appendix A). Of the several candidates of Ca^2+^-permeable channels in macrophages [5,23,24,25], the expression levels of the Orai1, TRPM2, and TRPM7 transcripts were high (Appendix A).

To examine whether K_Ca_3.1 expression levels are modified during the differentiation of M_0_ into M_2_ macrophages, the expression levels of K_Ca_3.1 in THP-1-derived M_0_ and M_2_ macrophages were measured using real-time PCR and Western blot assays. Those K_Ca_3.1 transcripts and proteins were almost the same between both groups (n = 4 for each, *p* > 0.05) (Figure 2C–E). Correspondingly, no significant difference in SKA-121-induced hyperpolarizing responses was found between both groups (*p* > 0.05) (Figure 2F). In addition, we measured 1 μM SKA-121-activated K_Ca_3.1 currents in THP-1-derived M_2_ macrophages by whole-cell patch-clamp recordings [26]. Cells were held at −80 mV and 200 msec ramps from −120 to +40 mV were delivered every 10 sec with a rate of 0.8 mV/ms. Figure 3A represents plots of current density versus voltage after the addition of vehicle (black), 1 μM SKA-121 (red), and 1 μM SKA-121 plus 1 μM TRAM-34 (blue). As shown in Figure 3B, SKA-121-sensitive K_Ca_3.1current components showed typical voltage independence and reversal potential of −71.9 ± 5.8 mV (n = 9). Outward currents at +40 mV were significantly increased by the addition of 1 μM SKA-121 (n = 9, *p* < 0.05 vs. vehicle control), and SKA-121-activated current components were reduced by the application of 1 μM TRAM-34 (n = 9, *p* < 0.01 vs. SKA-121-treated group) (Figure 3C).

### 2.3. Downregulation of IL-10 and IL-8 by the Treatment with SKA-121 in THP-1-Derived M_2_ Macrophages

We previously reported that the activation of K_Ca_3.1 with SKA-31 suppressed IL-10 expression and production in human T-cell lymphoma HuT-78 cells [27]. In this study, we examined the effects of treatment with SKA-121 (1 μM) for 24 h on the expression of IL-10, IL-8, VEGF-A, and TGF-β1 in THP-1-derived M_2_ macrophages (Figure 4). The expression levels of IL-10 and IL-8 transcripts were significantly suppressed by the SKA-121 treatment (n = 4 for each, *p* < 0.01), and simultaneous treatment with the K_Ca_3.1 blocker, TRAM-34 (10 μM), reversed this downregulation (Figure 4A,B). TRAM-34 alone did not elicit significant changes in the transcription of IL-10 and IL-8 (Appendix A). On the other hand, the expression of VEGF-A and TGF-β1 was not affected by the SKA-121 treatment (n = 4 for each, *p* > 0.05) (Figure 4C,D). Consistent with the results shown in Figure 4A,B, the secretion of IL-10 and IL-8 was decreased by the SKA-121 treatment (n = 4 for each, *p* < 0.01) (Figure 4E,F). These suggest that the K_Ca_3.1 activator suppressed the immunosuppressive and pro-tumorigenic functions of TAMs by repressing the transcription of IL-10 and IL-8.

### 2.4. Reversal Effects of Treatment with SKA-121 on High K^+^ Level-Induced Increases in IL-10 and IL-8 in THP-1-Derived M_2_ Macrophages

In a hypoxic TME, K^+^ released from necrotic cancer cells accumulates in the extracellular compartments [9]. An in vivo K^+^ imaging analysis by Tan et al. (2020) showed that K^+^ levels were elevated in the TME, with an average concentration of approximately 29 mM [28]. The concentration of K^+^ was 5 mM in the RPMI 1640 medium used in this study. Media with high K^+^ (final concentration: 25 mM) were prepared by KCl supplementation. Real-time PCR and ELISA assays showed the increased transcriptional expression (n = 4 for each, *p* < 0.01) (Figure 5A,B) and secretion (n = 4 for each, *p* < 0.01) (Figure 5C,D) of IL-10 and IL-8 in THP-1-derived M_2_ macrophages exposed to high [K^+^]_e_ (25 mM). Supplementation with 20 mM NaCl instead of 20 mM KCl did not affect their expression (n = 4 for each, *p* > 0.05); however, 10 mM K_2_HPO_4_ showed a similar effect to 20 mM KCl supplementation (n = 4 for each, *p* < 0.01) (Appendix A). 

We then investigated the effects of a 24 h treatment with SKA-121 on high [K^+^]_e_-induced increases in IL-8 and IL-10 in THP-1-derived M_2_ macrophages. High [K^+^]_e_-induced increases in the expression (Figure 5A,B) and secretion (Figure 5C,D) of IL-8 and IL-10 were not affected by simultaneous treatment with 1 μM SKA-121 (n = 4 for each, *p* > 0.05); however, was significantly reduced by simultaneous treatment with 10 μM SKA-121 (n = 4 for each, *p* < 0.01) (Figure 5). Neither the 24 h treatment with SKA-121 nor the exposure to high [K^+^]_e_ altered the viability of THP-1-derived M_2_ macrophages (n = 5 for each, *p* > 0.05) (Appendix A). These results demonstrated that the K_Ca_3.1 activator might suppress the overexpression of IL-10 and IL-8 in TAMs caused by increases in [K^+^]_e_ in the TME.

### 2.5. Involvement of ERK-CREB and/or JNK-c-Jun Cascades in the Transcriptional Repression of IL-10 and IL-8 in THP-1-Derived M_2_ Macrophages

Previous studies demonstrated that the following signaling pathways regulated the expression of IL-8 and IL-10: ERK/JNK/p38 mitogen-activated protein kinase (MAPK), PI3K/AKT/mTOR (mammalian target of rapamycin), NF–κB, and calcineurin/NFAT [29,30,31,32]. Among the 10 signaling pathway inhibitors examined, a 24 h treatment with the ERK1/2 inhibitor, SCH772984 (1 μM) markedly suppressed the expression levels of both IL-10 and IL-8 transcripts in THP-1-derived M_2_ macrophages (Figure 6A,D), which corresponded to reductions in their secretion by SCH772984 (Figure 6B,E). Additionally, treatment with the JNK inhibitor, SP600125 (10 μM), repressed the expression levels of both IL-10 and IL-8 transcripts (Figure 6A,D), which corresponded to decreases in their secretion by SP600125 (Figure 6C,F).

We then examined the effects of treatment with SKA-121 and exposure to high [K^+^]_e_ on phosphorylated ERK1/2 (P-ERK1/2), phosphorylated JNK (P-JNK), and phosphorylated c-Jun (P-c-Jun) levels in THP-1-derived M_2_ macrophages by Western blotting (Figure 7 and Figure 8). The ratio of P-ERK2 to total ERK2 in THP-1-derived M_2_ macrophages was decreased by the treatment with 1 μM SKA-121 (n = 4, *p* < 0.01) (Figure 7A,E), but was increased by the exposure to high [K^+^]_e_ (25 mM) (n = 4, *p* < 0.01) (Figure 7B,F). The very low intensity of the weak band signals for P-ERK1 and total ERK1 was observed. Additionally, the ratio of P-JNK to total JNK in THP-1-derived M_2_ macrophages was decreased by the SKA-121 treatment (n = 4, *p* < 0.01) (Figure 7C,G), but was increased by the exposure to high [K^+^]_e_ (n = 4, *p* < 0.01) (Figure 7D,H). Similarly, the ratio of P-c-Jun to total c-Jun was decreased by the SKA-121 treatment (n = 4, *p* < 0.01) (Figure 8A,C), but was increased by the exposure to high [K^+^]_e_ (n = 4, *p* < 0.01) (Figure 8B,D). These results suggested that the ERK and JNK-c-Jun signaling pathways might be, at least partly, mediated by the K_Ca_3.1 activation-induced transcriptional repression of IL-10 and/or IL-8 in THP-1-derived M_2_ macrophages.

Several transcription factors (i.e., CREB, AP-1, and/or NF–κB) that bind to the promoters of the IL-8 and IL-10 genes have been identified to date; however, the ERK1/2-CREB pathway is a candidate that is commonly associated with the transcription of IL-8 and IL-10 [19,20]. We, therefore, examined the effects of the potent and selective CREB inhibitor, 666-15 (1 μM), on the expression levels of IL-10 and IL-8 transcripts in THP-1-derived M_2_ macrophages (Figure 9). The expression and secretion of IL-10 and IL-8 were significantly decreased by the 24 h treatment with 666-15 (Figure 9A–D). The ratio of P-CREB to total CREB in THP-1-derived M_2_ macrophages was decreased by the treatment with 1 μM SKA-121 (n = 4, *p* < 0.01) (Figure 9E,G), but was increased by the exposure to high [K^+^]_e_ (25 mM) (n = 4, *p* < 0.01) (Figure 9F,H). Therefore, ERK-CREB and JNK-c-Jun cascades may be involved in K_Ca_3.1-regulated IL-10 and IL-8 in TAMs.

### 2.6. Suppressive Effects of Treatment with SKA-121 on the Up-Regulation of IL-10 and IL-8 in THP-1-Derived M_2_ Macrophages by the Exposure of Soluble Factors in Human Prostate Cancer PC-3 Cell-Cultured Media

Tumor-conditioned media are associated with the generation and infiltration of TAMs [33]. We examined the effects of exposure to human prostate cancer PC-3 spheroid-cultured media for 24 h on the expression levels of IL-10, IL-8, VEGF-A, and TGF-β1 in THP-1-derived M_2_ macrophages. Media were prepared by the 3D spheroid formation of PC-3 cells for 7 days using ultra-low attachment surface coating cultureware. As shown in Figure 10A,B (left columns), IL-10 and IL-8 expression levels in THP-1-derived M_2_ macrophages were increased by approximately 3-fold following the exposure to PC-3 media. Simultaneous treatment with 1 or 10 μM SKA-121 resulted in significant decreases in the expression levels of both transcripts (n = 4 for each, *p* < 0.01) (Figure 10A,B) without changes in those of VEGF-A or TGF-β1 (n = 4 for each, *p* > 0.05) (Figure 10C,D). Furthermore, elevated IL-10 and IL-8 secretion levels by approximately 1.5-fold following the exposure to PC-3 media were significantly decreased by simultaneous treatment with SKA-121 (n = 4, *p* < 0.01) (Figure 10E,F). The 24 h exposure to PC-3 media did not alter the viability of THP-1-derived M_2_ macrophages (n = 5 for each, *p* > 0.05) (Appendix A).

## 3. Discussion

The intermediate-conductance Ca^2+^-activated K^+^ channel, K_Ca_3.1, is a key regulator of the polarization and tumor infiltration of TAMs. In the TME, tumor-infiltrating, M_2_-differentiated TAMs promote escape from cancer immune surveillance, thereby enhancing invasive and metastatic potentials and angiogenesis in cancer [1,2]. The various stresses preventing antitumor immune responses in the TME, such as hypoxia, high lactate levels, and high [K^+^]_e_ levels, affect TAM functions [34]. Accumulated K^+^ in the TME overcomes antitumor immune surveillance by inhibiting cytotoxic lymphocyte functions [9,10]. K^+^ channels as an ‘ionic checkpoint’ are the potential targets for cancer immunotherapy because activators of these channels, specifically expressing immune cells, may overcome the evasion of immune surveillance. In this study, we demonstrated that selective K_Ca_3.1 activators can potentially treat increased TAM-mediated cancer metastatic potential and angiogenesis in the TME using THP-1-derived M_2_ macrophages. The main results of this study are as follows: (1) pro-tumorigenic IL-8 and IL-10 expression and secretion in THP-1-derived M_2_ macrophages were significantly decreased by treatment with a K_Ca_3.1 activator (Figure 4), (2) high [K^+^]_e_ level-induced increases in IL-8 and IL-10 levels were reversed by treatment with a K_Ca_3.1 activator (Figure 5), (3) the transcription of IL-10 and IL-8 was promoted through the ERK-CREB and/or JNK-c-Jun cascades with increases in their phosphorylation levels (Figure 6, Figure 7, Figure 8 and Figure 9), and (4) prostate cancer cell-cultured medium-induced increases in IL-10 and IL-8 levels were reversed by treatment with a K_Ca_3.1 activator (Figure 10). Alternatively, the treatment with a K_Ca_3.1 activator did not affect the expression levels of VEGF-A or TGF-β1, which promote the immunosuppressive activity of T_reg_ cells and their recruitment to the TME (Figure 4C,D). We previously reported K_Ca_3.1 activator-induced reductions in the expression and secretion of IL-10 in IL-10-producing T-cell lymphoma, HuT-78 cells [27]. In contrast to native T_reg_ cells naturally present in the immune system, the NF–κB signaling pathway is continuously active in HuT-78 cells, and IL-10 transcription is regulated through the TGF-β-mediated Smad2/3 signaling pathway [27]. However, as shown in Figure 6A,D, the TGF-β signaling pathway did not contribute to the transcription of IL-10 or IL-8 in THP-1-derived M_2_ macrophage. No changes in the expression levels of IL-10 and IL-8 were found by the treatment with 10 ng/mL TGF-β1 for 24 h [0.0342 ± 0.0002 and 0.0336 ± 0.0009 (in arbitrary units) in vehicle- and TGF-β-treated groups, respectively, n = 4 for each, *p* > 0.05].

TAMs play a pivotal role in impairing the functions of antitumor, cytotoxic CD8^+^ T cells, and natural killer cells by releasing the anti-inflammatory cytokine IL-10 [1,4]. Multiple distinct signaling pathways were found to be responsible for the transcriptional regulation of IL-10 in immune and non-immune cells [35,36]. The IL-10 promoter was activated by CREB through the ERK1/2 signaling pathway in native THP-1 cells [37]. Correspondingly, this study showed the transcriptional repression of IL-10 in THP-1-derived M_2_ macrophages by a single treatment with an ERK1/2 inhibitor or a CREB inhibitor (Figure 6A and Figure 9A) and the reduced phosphorylation levels of ERK1/2 and CREB in THP-1-derived M_2_ macrophages treated with a K_Ca_3.1 activator (Figure 7A,E and Figure 9E,G). We recently reported that the inhibition of K_Ca_3.1 upregulated the expression of IL-10 through the JNK/c-Jun cascade in T_reg_ cells [38]. In this study, the JNK/c-Jun cascade was involved in the K_Ca_3.1 activation-induced down-regulation of IL-10 (Figure Figure 7C,G and Figure 8A,C). Collectively, the present results represent the first time that K_Ca_3.1 activators are a possible target for overcoming IL-10-mediated escape from tumor immune surveillance in the TME, through their inhibition of the IL-10 in TAMs via the ERK-CREB and JNK-c-Jun cascades.

Similar to IL-10, IL-8 secreted from TAMs plays a critical role in promoting tumor immunosuppression [39]. IL-8 has also been shown to play a role in the establishment of cancer stemness, chemoresistance, and tumor neovascularization in the TME [39,40]. In this study, the K_Ca_3.1 activator suppressed the production of IL-8 by repressing its transcription in THP-1-derived M_2_ macrophages (Figure 4B,F). Importantly, the K_Ca_3.1 activator also suppressed high [K^+^]_e_ exposure-induced IL-8 overexpression (Figure 5B,D). These results suggest that the down-regulation of IL-8 by the K_Ca_3.1 activator in TAMs not only prevented cancer migration and metastasis but also reduced microvessel density in cancer. However, further studies will be needed to prove this hypothesis in the future. Additionally, IL-8 transcription was distinctly repressed by ERK and JNK inhibitors in THP-1-derived M_2_ macrophages (Figure 6D). Correspondingly, IL-8 transcription was promoted by the ERK and JNK signaling pathways with their phosphorylation in several epithelial cells and synovial fibroblasts [18,41]. As shown in Figure 7, Figure 8 and Figure 9, the treatment with SKA-121 reduced the phosphorylation levels of ERK1/2, JNK, c-Jun, and CREB in THP-1-derived M_2_ macrophages, while the exposure to high [K^+^]_e_ exerted the opposite effects. These results suggest that the mechanisms underlying the K_Ca_3.1 activator-induced down-regulation and high [K^+^]_e_-induced upregulation of IL-8 and IL-10 are mediated via the same signaling pathways.

Angiogenic inhibition therapy is a promising strategy for cancer. TAMs secrete angiogenic factors such as VEGF-A and TGF-β1; however, the K_Ca_3.1 activator did not alter their expression in this study (Figure 4C,D). IL-8 cooperates with VEGF-A to promote tumor neovascularization [39]. Therefore, K_Ca_3.1 activators may be a novel strategy to terminate/initiate tumor neovascularization by reducing IL-8 secreted from tumor-infiltrating TAMs. Intertumoral IL-8 leads to the recruitment and accumulation of TAMs in the TME [39,40,42]. Bilaterally, IL-8 secreted from TAMs contributes to the epithelial-mesenchymal transition (EMT) process with cancer stem cell properties such as chemoresistance in the TME via a paracrine pathway [42,43]. In this study, IL-8 and IL-10 mRNA expression and secretion levels in THP-1-derived M_2_ macrophages increased in the culture media of prostate cancer PC-3 cells (Figure 10A,B,E,F). Bidirectional crosstalk between TAMs and cancer cells promotes TAMs to secrete IL-8 and IL-10 into the TME, resulting in the tumorigenesis, metastasis, and angiogenesis of solid cancers. A recent study indicated that exosomal microRNA from cancer cells promoted the M_2_ polarization of TAMs [44]. miR-21 mimicked increases in IL-10 mRNA levels and a miR-21 inhibitor prevented this alternation [44]. Indeed, several studies have shown that miR-21 is highly expressed in PC-3 cells and is upregulated by the spheroid formation and EMT of PC-3 cells [45,46]. Therefore, the PC-3-medium-induced upregulation of IL-10 and IL-8 in TAMs may be attributed to microRNA(s) in cancer cells, promoting EMT in the TME.

Feng et al. (2018) reported that the activation of Nrf2 upregulated the expression of M_2_ markers (CD163 and Arg1) [47]. Therefore, the upregulation of IL-10 and IL-8 following the M_2_ differentiation of THP-1 cells may be due to the activation of Nrf2. The expression levels of Nrf2 were high in THP-1-derived M_2_ macrophages (Appendix A); correspondingly, the transcriptional and secretory levels of IL-10 and IL-8 were significantly decreased by the 24 h treatment with ML385 (n = 4 for each, *p* < 0.01) (Appendix A–E). These results suggest that Nrf2 is involved in the upregulation of IL-10 and IL-8 but not in the SKA-121-induced down-regulation of them in THP-1-derived M_2_ macrophages (Appendix A). CCL22, which is secreted from TAMs, is responsible for the recruitment of T_reg_ cells to tumors, and the prevention of cytotoxic CD8^+^ T cell activation. No significant changes were observed in the expression levels of CCL22 transcripts following the 24 h treatment with SKA-121 (Appendix A).

## 4. Materials and Methods

### 4.1. Materials and Reagents

RPMI 1640 medium was purchased from FUJIFILM Wako Pure Chemicals (Osaka, Japan). Fetal bovine serum was from Sigma-Aldrich (St. Louis, MO, USA). ECL advanced chemiluminescence reagents were from Nacalai Tesque (Kyoto, Japan). Primary antibodies against K_Ca_3.1 (rabbit polyclonal), ERK1/2 (rabbit polyclonal), phospho-ERK1(T202/Y204)/ERK2(T185/Y187) (rabbit monoclonal), JNK (rabbit polyclonal), phospho-JNK(Y185) (rabbit polyclonal), c-Jun (rabbit polyclonal), CREB (rabbit polyclonal), phospho-CREB(S133) (rabbit polyclonal), and β-actin (ACTB) (mouse monoclonal) were from Alomone Labs (Jerusalem, Israel), BioLegend (San Diego, CA, USA), R&D Systems (Minneapolis, MN, USA), GeneTex (Alton Pkwy Irvine, CA, USA), ProteinTech (Rosemont, IL, USA), Medical & Biological Laboratories (Nagoya, Japan), and ABclonal (Tokyo, Japan). Horseradish peroxidase (HRP)-conjugated anti-rabbit and mouse HRP-conjugated IgG secondary antibodies were from Merck (Darmstadt, Germany). Recombinant human IL-4 and IL-13 were from PeproTech (Cranbury, NJ, USA). TRAM-34 (Santa Cruz Biotechnology, Santa Cruz, CA, USA), SKA-121 [48] (MedChemExpress), Luna Universal qPCR master Mix (New England Biolabs Japan, Tokyo, Japan), the IL-10/IL-8/IL-1β Human Uncoated ELISA kits (Thermo Fisher Scientific, Waltham, MA, USA), AZD5363, LY364947, everolimus, SCH772984, Bay11-7082 (Cayman Chemical, Ann Arbor, MI, USA), LY294002 (Chem Scene, Monmouth Junction, NJ, USA), ciclosporin A (FUJIFILM Wako Pure Chemicals), T-5224 (APExBIO, Boston, MA, USA), ML385, sulforaphane, 666-15 (Selleckchem), DiBAC_4_(3), Fura 2-acetoxymethyl ester, WST-1, 1-methoxy PMS (Dojindo, Kumamoto, Japan), SP600125 (LC Laboratories, Woburn, MA, USA), and phorbol 12-myristrate 13-acetate (PMA) (AdipoGen, San Diego, CA, USA) were also purchased form the indicated sources. Other chemicals and reagents were from Sigma-Aldrich, FUJIFILM Wako Pure Chemicals, and Nacalai Tesque.

### 4.2. Cell Culture and Differentiation into M_2_ Macrophages

The differentiation of the human monocytic leukemia cell line, THP-1, into M_0_ macrophages was induced by treatment with PMA treatment (100 ng/mL) for 24 h. After removal of the medium, cells were incubated with Roswell Park Memorial Institute (RPMI) 1640 medium supplemented with IL-4 and IL-13 (20 ng/mL each) for 72 h to induce the polarization of M_2_ macrophages. Drug applications were performed 48 h after the incubation with IL-4/IL-13 [3].

### 4.3. Measurements of the Membrane Potential and [Ca^2+^]_i_

The membrane potential was measured using the fluorescent voltage-sensitive dye, DiBAC_4_(3), as previously reported [27]. In membrane potential imaging, cells loaded with DiBAC_4_(3) were illuminated at a wavelength of 490 nm. [Ca^2+^]_i_ was measured using the fluorescent Ca^2+^ indicator dye, Fura 2-AM, and cells loaded with Fura 2 were alternatively illuminated at wavelengths of 340 and 380 nm. Before the fluorescence measurements, cells were incubated in the extracellular solution containing 100 nM DiBAC_4_(3) and 10 μM Fura 2-AM for 20 min. The extracellular solution was composed of (in mM) the following: 137 NaCl, 5.9 KCl, 2.2 CaCl_2_, 1.2 MgCl_2_, 14 glucose, and 10 HEPES [4-(2-hydroxyethyl)piperazine-1-ethanesulfonic acid] (pH 7.4). A Ca^2+^-free solution was prepared by replacing 2.2 mM CaCl_2_ with 1 mM EGTA [ethylene glycol bis(beta-aminoethylether)-N,N,N,N-tetraacetic acid]. Fluorescence images were recorded using the ORCA-Flash2.8 digital camera (Hamamatsu Photonics, Hamamatsu, Japan). Data collection and analyses were performed using an HCImage system (Hamamatsu Photonics). Images were obtained every 5 sec, and fluorescent intensity values were assessed using the average for 1 min (12 images). The fluorescent intensity of Fura 2 was expressed as measured 340/380 nm fluorescence ratios after background subtraction. Drug solutions (SKA-121 alone and SKA-121 plus TRAM-34) were applied by operating the three-way stopcock.

### 4.4. Whole-Cell Patch-Clamp Recording

Whole-cell patch-clamp recording experiments were performed as reported previously [26]. A whole-cell patch-clamp was applied to single THP-1-derived M_2_ macrophages using an Axopatch 200B patch-clamp amplifier under the control of pClamp 11 software (Molecular Devices, San Jose, CA, USA) (23 ± 1 °C). Data acquisition and analysis of whole-cell currents were performed using Clampfit software (Molecular Devices). The resistance of microelectrodes filled with pipette solution was 3–5 MΩ. A ramp voltage protocol from −120 mV to + 40 mV for 200 msec was applied every 10 s at a holding potential of −80 mV. The external solution was (in mM): 137 NaCl, 5.9 KCl, 2.2 CaCl_2_, 1.2 MgCl_2_, 14 glucose and 10 HEPES, pH 7.4. The pipette solution was (in mM): 140 KCl, 4 MgCl_2_, 3.16 CaCl_2_, 5 EGTA, 10 HEPES and 2 Na_2_ATP, pH 7.2, with an estimated free Ca^2+^ concentration of 300 nM (pCa 6.5) [26].

### 4.5. Preparation of Cancer Spheroid Models Using Ultra-Low Attachment Surface Coating-Cultureware

The human prostate cancer cell line, PC-3, was purchased from the RIKEN Cell Bank (Osaka, Japan). Cells were cultured in RPMI 1640 medium supplemented with 10% fetal bovine serum and Penicillin-Streptomycin Mixture (FUJIFILM Wako Pure Chemicals). All cells were cultured in a humidified atmosphere containing 5% CO_2_ at 37 °C. The PrimeSurface^®^ system (Sumitomo Bakelite, Tokyo, Japan) was used for three-dimensional spheroid formation. Cell suspensions were seeded onto a PrimeSurface 96U plate at 10^4^ cells/well, and then cultured for 7 days.

### 4.6. Real-Time PCR

As previously reported, total RNA extraction and cDNA synthesis were conducted [8]. The gene-specific primers for real-time PCR examinations were designed using Primer Express^TM^ software (Ver 1.5, Thermo Fisher Scientific). Real-time PCR was performed using Luna Universal qPCR Master Mix (New England Biolabs Japan, Tokyo, Japan) on the ABI 7500 real-time PCR instrument (Applied Biosystems) [8]. The following PCR primers were used: IL-8 (GenBank accession number: BC013615, 218–337, 120 bp); IL-10 (NM_000572, 430–549, 120 bp); VEGF-A (NM_001025366, 1085–1204, 120 bp); TGF-β1 (NM_000660, 1211–1331, 120 bp); CD163 (NM_004244, 3214–3333, 120 bp); arginase 1 (Arg1) (NM_001244438, 711–820, 110 bp); CCL22 (NM_002990, 175–274, 100 bp); K_Ca_3.1 (NM_002250, 1191–1311, 121 bp); Nrf2 (NM_006164, 1704–1823, 120 bp); Orai1 (NM_032790.3, 926–1046, 121 bp); TRPV2 (NM_016113, 1840–1959, 120 bp); TRPM2 (NM_003307, 3082–3224, 143 bp) TRPM7 (NM_017672.6, 2167–2286, 120 bp); Piezo1 (NM_001142864.4, 7510–7629, 120 bp), and ACTB (NM_001101, 411–511, 101 bp). Unknown quantities relative to the standard curve for a particular set of primers were calculated [8], yielding the transcriptional quantitation of gene products relative to ACTB.

### 4.7. Western Blots

Whole-cell lysates were extracted by RIPA buffer. Halt^TM^ phosphatase inhibitor cocktail (1×) (Thermo Fisher Scientific) was added to extracts at a final concentration of 1%. Equal amounts of protein were subjected to SDS-PAGE and immunoblotting with anti-P-ERK1/2 polyclonal (rabbit) (1:1500) (42/44 kDa), anti-ERK1/2 polyclonal (rabbit) (1:3000) (42/44 kDa), anti-P-JNK polyclonal (rabbit) (1:1500) (43/50 kDa), anti-JNK polyclonal (rabbit) (1:2000) (43/50 kDa), anti-c-Jun polyclonal (rabbit) (1:2000) (42–46 kDa), anti-CREB polyclonal (rabbit) (1:2000) (45 Da), anti-P-CREB polyclonal (rabbit) (1:2000) (45 kDa), anti-K_Ca_3.1 polyclonal (rabbit) (50 kDa), and anti-ACTB monoclonal (mouse) (1:15,000) (43 kDa) antibodies, and were then incubated with anti-rabbit or anti-mouse HRP-conjugated IgG secondary antibody. An ECL advance chemiluminescence reagent kit (Nacalai Tesque) was used to identify the bound antibody. The resulting images were analyzed using Amersham Imager 600 (GE Healthcare Japan) [38]. The optical density of the protein band signal relative to that of the ACTB signal was calculated using ImageJ software (Ver. 1.42, NIH, USA), and protein expression levels in the vehicle control were then expressed as 1.0.

### 4.8. Measurement of Cytokine Production by Enzyme-Linked Immunosorbent Assay (ELISA)

Human IL-10 and IL-8 levels in supernatant samples were measured with the respective IL-10/IL-8 Human Uncoated ELISA kits (Thermo Fisher Scientific), according to the manufacturer’s protocols. Standard curves were plotted using a series of cytokine/chemokine concentrations.

### 4.9. Statistical Analysis

Statistical analyses were performed with the statistical software XLSTAT (version 2013.1). The unpaired/paired Student’s *t*-test with Welch’s correction or one-way ANOVA with Tukey’s test was used to assess the significance of differences between two groups and among multiple groups. Results with a *p*-value < 0.05 were considered to be significant. Data are shown as means ± standard error.

## 5. Conclusions

In this study, elevated [K^+^]_e_ by necrotic cancer and cancer-infiltrating cells in the TME increased the expression and production of IL-10 and IL-8 in THP-1-derived M_2_ macrophages. We showed the effectiveness of K_Ca_3.1 activators for TME-mediated TAM dysregulation, generating the overexpression of IL-10 and IL-8. Several potent and selective K_Ca_3.1 activators, which positively modulate channel-gating, have been developed [48]; however, their clinical and therapeutic importance remains unclear. The present results provide novel insights into recent advances in TAM targeting therapies as antitumor strategies and indicate that potent and selective K_Ca_3.1 activators need to be considered for future therapeutic applications in cancer immunotherapy. Further studies under the TME-mimicking conditions, such as acidic pH, hypoxia, high lactate, low glucose, and so on will be needed to evaluate the clinical applications and limitations K_Ca_3.1 activators in cancer immunotherapy.

## Figures and Tables

**Figure 1 ijms-23-08603-f001:**
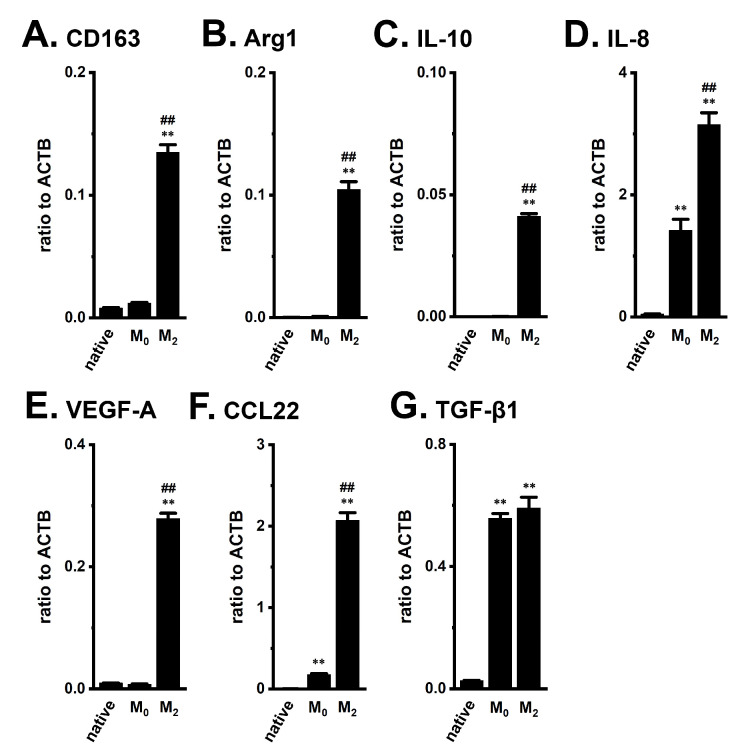
Gene expression of M_2_ markers, cytokines, chemokines, and growth factors in THP-1-derived M_2_-like macrophages. **A**–**G**: Real-time PCR examination of CD163 (**A**), Arg1 (**B**), IL-10 (**C**), IL-8 (**D**), VEGF-A (**E**), CCL22 (**F**), and TGF-β1 (**G**) expression in native THP-1 (‘native’), THP-1-derived M_0_-like macrophages (‘M_0_’), and THP-1-derived M_2_-like macrophages (‘M_2_’). Expression levels are shown as a ratio to ACTB (n = 4 for each). **: *p* < 0.01 vs. ‘native’, ^##^: *p* < 0.01 vs. ‘M_0_’.

**Figure 2 ijms-23-08603-f002:**
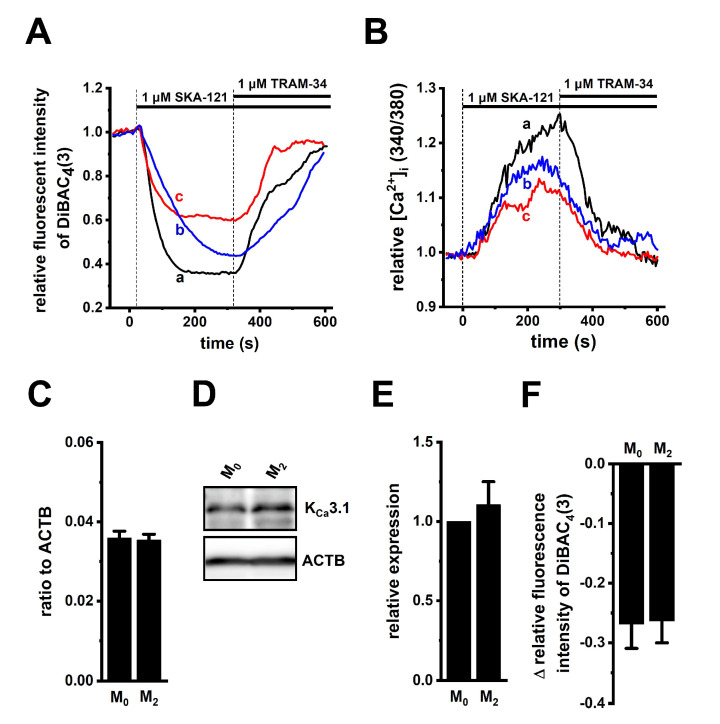
Functional expression of K_Ca_3.1 in THP-1-derived M_2_ macrophages. (**A**,**B**): Simultaneous measurements of changes in the membrane potential (**A**) and [Ca^2+^]_i_ (**B**) in the three different cells [a (black), b (blue), and c (red)], following the application of a selective K_Ca_3.1 activator, SKA-121 (1 μM), and/or the K_Ca_3.1 inhibitor, TRAM-34 (1 μM) using DiBAC_4_(3) and Fura 2, respectively. The relative time-course changes in fluorescence intensities (1.0 at 0 s) from three different THP-1-derived M_2_ macrophages are shown. (**C**): Real-time PCR examination of K_Ca_3.1 expression in THP-1-derived M_0_ macrophages (‘M_0_’) and M_2_ macrophages (‘M_2_’). Expression levels are shown as a ratio to ACTB (*p* > 0.05, n = 4 for each). (**D**,**E**): Protein expression levels of K_Ca_3.1 in the ‘M_0_’ and ‘M_2_’ groups were determined by Western blot. Specific band signals for K_Ca_3.1 were observed at approximately 50 kDa (**D**, upper panel). After compensation with the optical density of the ACTB signal (43 kDa) (**D**, lower panel), the expression level in the ‘M_0_’ group was expressed as 1.0 (*p* > 0.05, n = 4 for each) (**E**). (**F**): SKA-121 (1 μM)-induced relative hyperpolarizing responses in the ‘M_0_’ and ‘M_2_’ groups (*p* > 0.05, n = 37 and 29, respectively).

**Figure 3 ijms-23-08603-f003:**
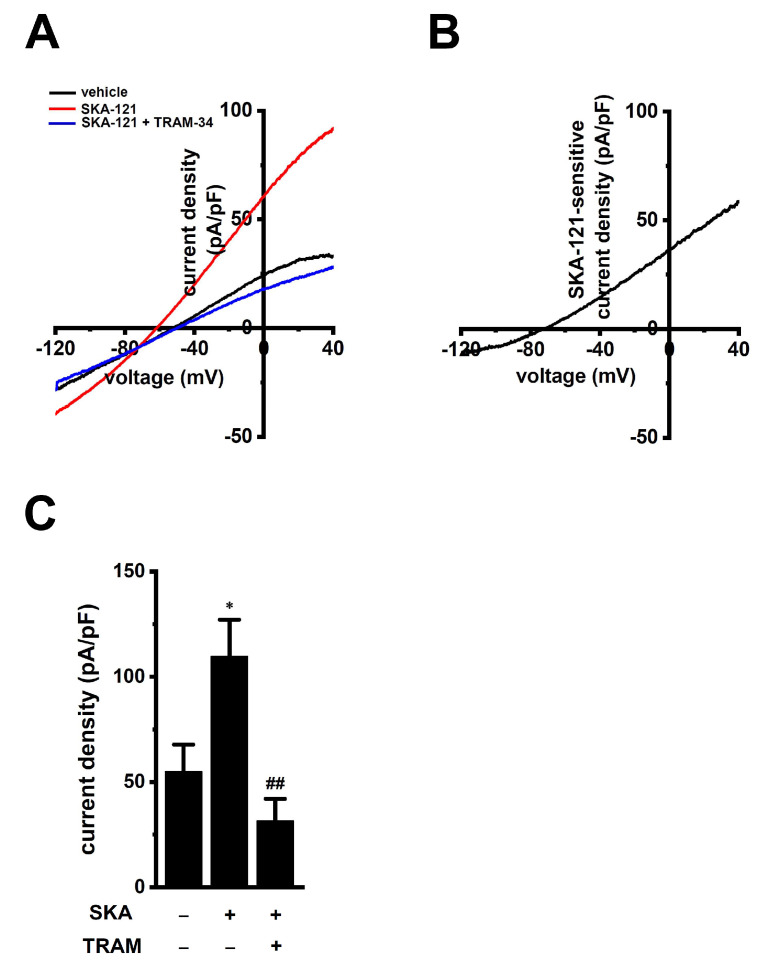
Whole-cell patch-clamp recordings of SKA-121-activated K^+^ currents in THP-1-derived M_2_ macrophages. (**A**): Typical current density and voltage-relationships after addition of vehicle (black), 1 μM SKA-121 (red), and 1 μM SKA-121 plus 1 μM TRAM-34 (blue). Currents were elicited by ramp depolarization from −120 to +40 mV from a holding potential of −80 mV every 10 s. (**B**): Typical current density and voltage-relationship of SKA-121-sensitive component. (**C**): Summarized results of current densities (pA/pF) at +40 mV in three groups (n = 9 for each). *: *p* < 0.05 vs. vehicle control (−/−); ^##^
*p* < 0.01 vs. SKA-121 alone (+/−).

**Figure 4 ijms-23-08603-f004:**
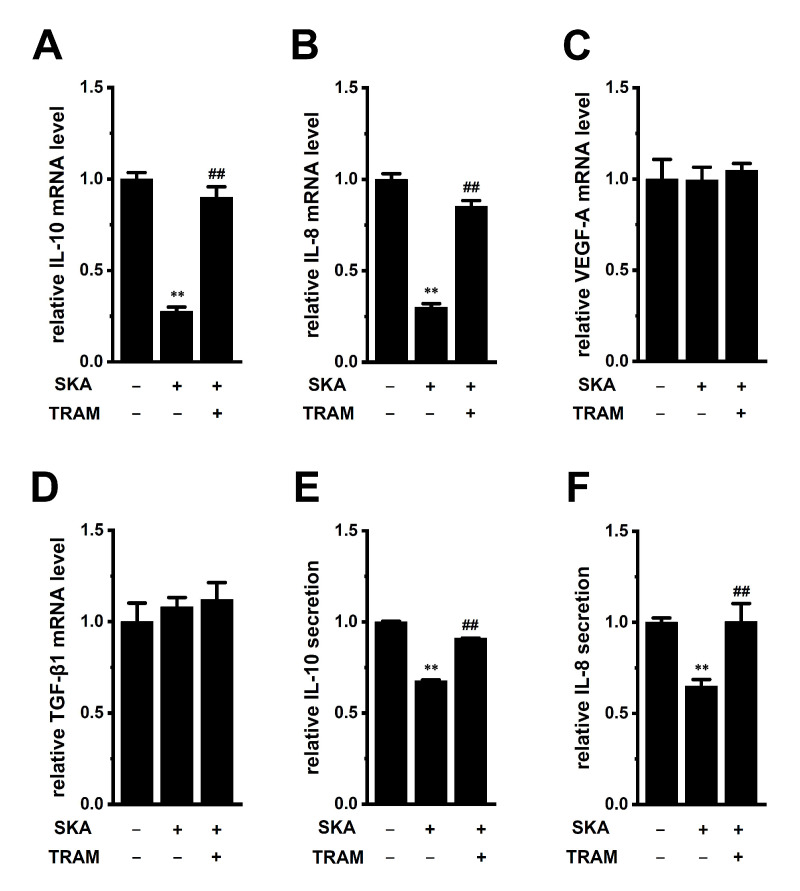
Effects of treatment with SKA-121 on IL-10, IL-8, VEGF-A, and TGF-β1 expression and on IL-10 and IL-8 secretion in THP-1-derived M_2_ macrophages. A–D: Real-time PCR examination of IL-10 (**A**), IL-8 (**B**), VEGF-A (**C**), and TGF-β1 (**D**) expression in THP-1-derived M_2_ macrophages treated (+) or untreated (−) with 1 μM SKA-121 and 10 μM TRAM-34 for 24 h. Relative mRNA expression in the vehicle control (‘−’ for SKA and ‘−’ for TRAM) is expressed as 1.0 (n = 4 for each). (**E**,**F**): Quantitative detection of IL-10 (**E**) and IL-8 (**F**) secretion by an ELISA assay in THP-1-derived M_2_ macrophages treated and untreated with SKA-121 and TRAM-34. Relative cytokine secretion in the vehicle control (−/−) is expressed as 1.0 (n = 4 for each). **: *p* < 0.01 vs. the vehicle control (−/−), ^##^: *p* < 0.01 vs. SKA-121 alone (+/−).

**Figure 5 ijms-23-08603-f005:**
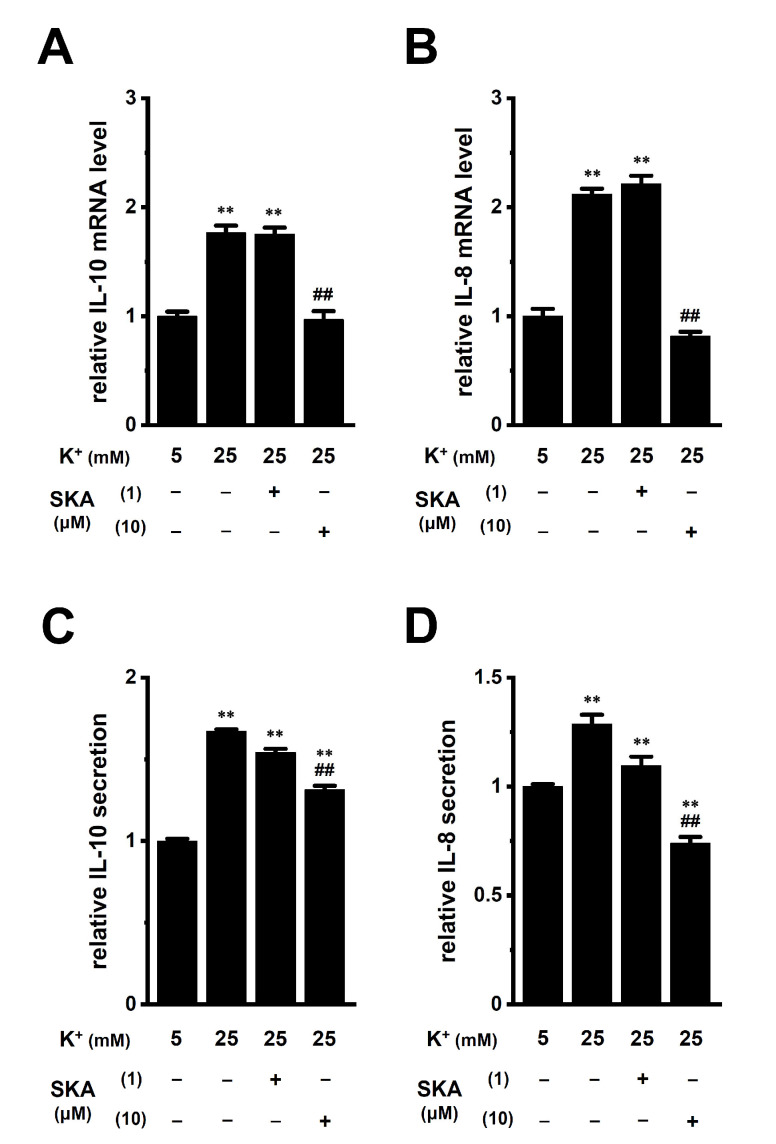
Effects of treatment with SKA-121 on high [K^+^]_e_-enhanced IL-10 and IL-8 expression and secretion in THP-1-derived M_2_ macrophages. **A**,**B**: Real-time PCR examination of IL-10 (**A**) and IL-8 (**B**) expression in normal [K^+^]_e_ (5 mM)- and high [K^+^]_e_ (25 mM)-treated THP-1-derived M_2_ macrophages for 24 h in the presence (+) or absence (−) of SKA-121 (1, 10 μM). Relative mRNA expression in normal [K^+^]_e_ is expressed as 1.0 (n = 4 for each). (**C**,**D**): Quantitative detection of IL-10 (**C**) and IL-8 (**D**) secretion by an ELISA assay in THP-1-derived M_2_ macrophages treated and untreated with SKA-121. Relative cytokine secretion in normal [K^+^]_e_ is expressed as 1.0 (n = 4 for each). **: *p* < 0.01 vs. normal [K^+^]_e_, ^##^: *p* < 0.01 vs. the vehicle control (−/−) of high [K^+^]_e_.

**Figure 6 ijms-23-08603-f006:**
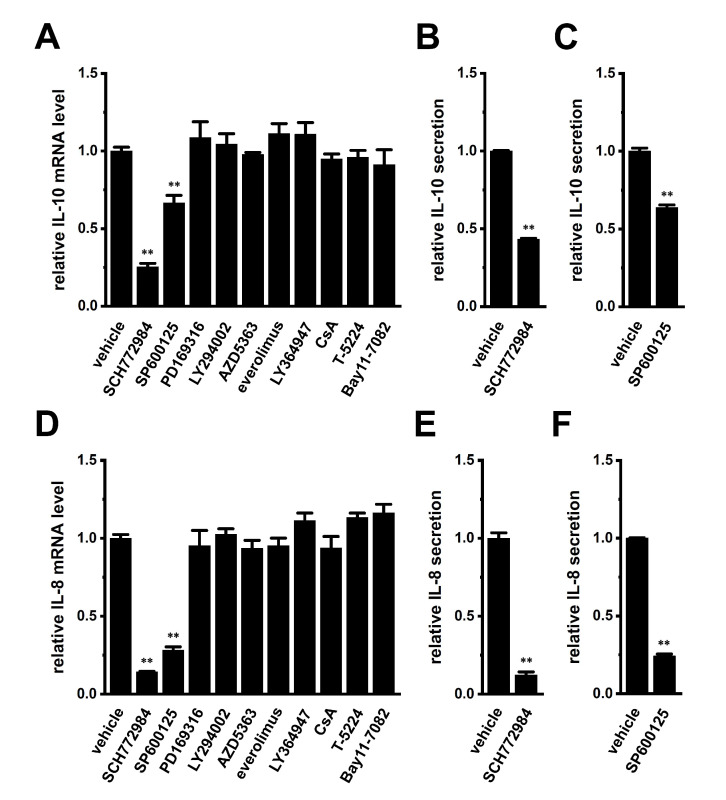
Effects of a 24 h treatment with various signaling pathway inhibitors on IL-10 and IL-8 expression and effects of the 24 h treatment with the ERK1/2 inhibitor, SCH772984 and JNK inhibitor, SP600125 on IL-10 and IL-8 secretion in THP-1-derived M_2_ macrophages. (**A**,**D**): Real-time PCR examination of IL-10 (**A**) and IL-8 (**D**) expression in SCH772964 (1 μM)-, SP600125 (10 μM)-, PD169316 (10 μM)-, LY294002 (10 μM)-, AZD5363 (2 μM)-, everolimus (10 nM)-, LY364947 (10 μM)-, ciclosporin A (CsA, 1 μM)-, T-5224 (10 μM)-, and Bay11-7082 (10 μM)-treated THP-1-derived M_2_ macrophages for 24 h. Relative mRNA expression in the vehicle control is expressed as 1.0 (n = 4 for each). (**B**,**C**,**E**,**F**): Quantitative detection of IL-10 (**B**,**C**) and/or IL-8 (**E**,**F**) secretion by an ELISA assay in SCH772964 (**B**,**E**)- and SP600125 (**C**,**F**)-treated THP-1-derived M_2_ macrophages. Relative cytokine secretion in the vehicle control is expressed as 1.0 (n = 4 for each). **: *p* < 0.01 vs. the vehicle control.

**Figure 7 ijms-23-08603-f007:**
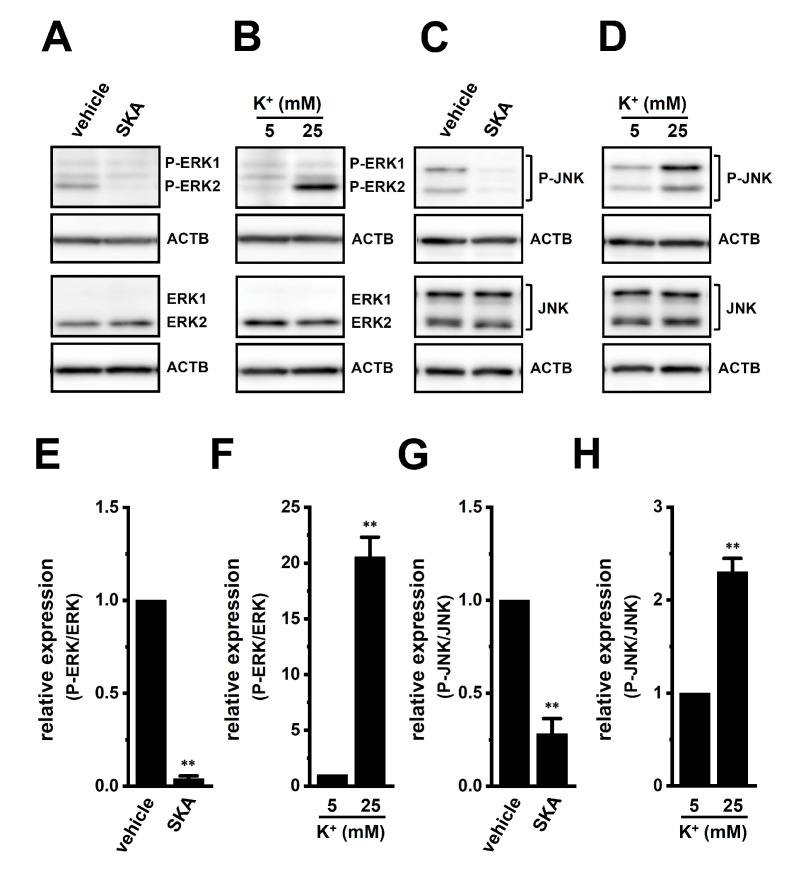
Protein expression levels of phosphorylated ERK1/2 (P-ERK1/2) and P-JNK in THP-1-derived M_2_ macrophages following the SKA-121 treatment and high [K^+^]_e_ exposure. **A**–**D**: Western blot showing P-ERK1/2, total ERK1/2 (ERK1/2) (**A,B**), P-JNK, and total JNK (JNK) (**C**,**D**) in 1 μM SKA-121-treated (**A**,**C**) and high [K^+^]_e_-exposed (**B**,**D**) THP-1-derived M_2_ macrophages. Specific band signals were observed at 42 (P-ERK2), 42 (ERK2), 43/50 (P-JNK), and 43/50 (JNK) kDa. **E**–**H**: Summarized results of the relative protein expression of P-ERK2/ERK2 (**E**,**F**) and P-JNK/JNK (**G**,**H**) in 1 μM SKA-121-treated (**E**,**G**) and high [K^+^]_e_-exposed (**F**,**H**) THP-1-derived M_2_ macrophages. After compensation with the optical density of the ACTB signal (43 kDa), the expression level in the vehicle control or 5 mM K^+^ is expressed as 1.0 (n = 4 for each). **: *p* < 0.01 vs. the vehicle control and 5 mM K^+^.

**Figure 8 ijms-23-08603-f008:**
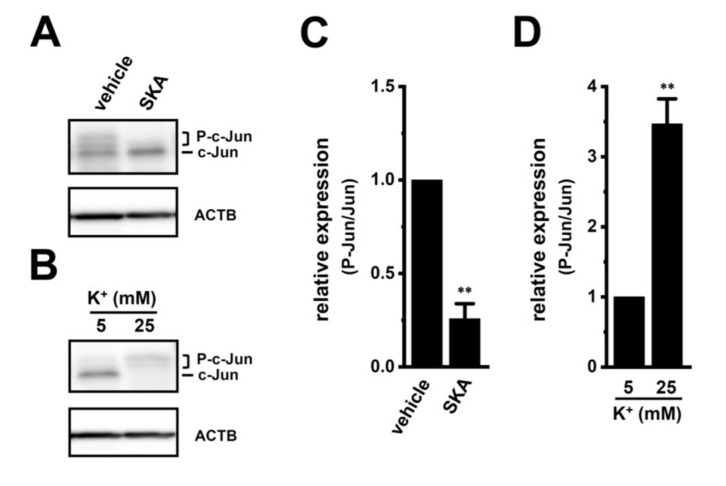
Protein expression levels of phosphorylated c-Jun (P-c-Jun) following the SKA-121 treatment and high [K^+^]_e_ exposure in THP-1-derived M_2_ macrophages. (**A**,**B**): Western blot showing P-c-Jun, and unphosphorylated c-Jun (c-Jun) in 1 μM SKA-121-treated (**A**) and high [K^+^]_e_-exposed (**B**) THP-1-derived M_2_ macrophages. Specific band signals were observed at 42-46 (P-c-Jun) and 40 (c-Jun) kDa. (**C**,**D**): Summarized results of the relative protein expression of P-c-Jun/c-Jun in 1 μM SKA-121-treated (**C**) and high [K^+^]_e_-exposed (**D**) THP-1-derived M_2_ macrophages. After compensation with the optical density of the ACTB signal (43 kDa), the expression level in the vehicle control is expressed as 1.0 (n = 4 for each). **: *p* < 0.01 vs. the vehicle control and 5 mM K^+^.

**Figure 9 ijms-23-08603-f009:**
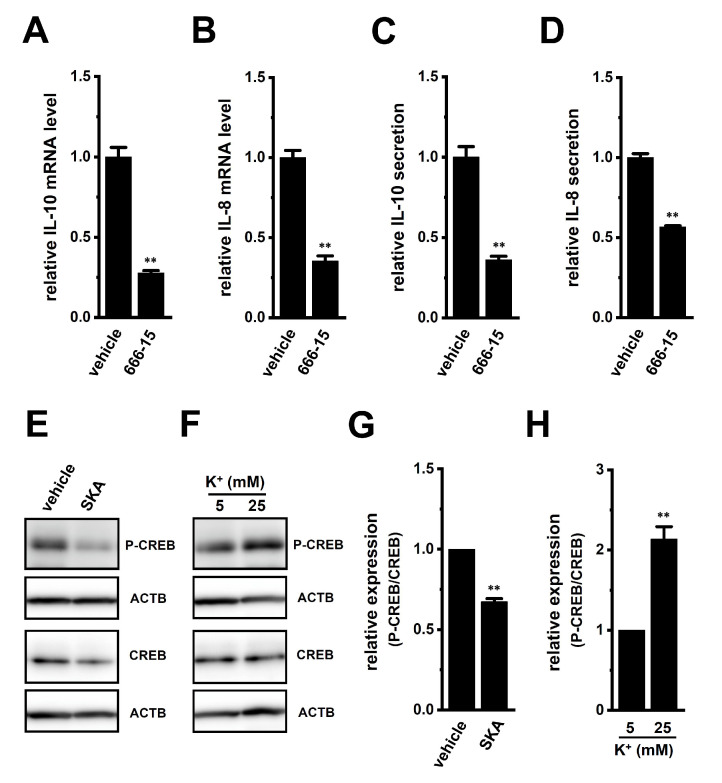
Effects of treatment with a CREB inhibitor on expression levels of IL-10, IL-8, VEGF-A, and TGF-β1 transcripts and on phosphorylated P-CREB protein levels following the SKA-121 treatment and high [K^+^]_e_ exposure in THP-1-derived M_2_ macrophages. (**A**–**D**): Real-time PCR examination of IL-10 (**A**), IL-8 (**B**), VEGF-A (**C**), and TGF-β1 (**D**) expression in THP-1-derived M_2_ macrophages treated with vehicle and 1 μM 666-15 for 24 h. Relative mRNA expression in the vehicle control is expressed as 1.0 (n = 4 for each). E-F: Western blot showing P-CREB and total CREB (CREB) in 1 μM SKA-121-treated (**E**) and high [K^+^]_e_-exposed (**F**) THP-1-derived M_2_ macrophages. Specific band signals for P-CREB and CREB were observed at approximately 40 kDa. G, H: Summarized results of the relative protein expression of P-CREB/CREB in 1 μM SKA-121-treated (**G**) and high [K^+^]_e_-exposed (**H**) THP-1-derived M_2_ macrophages. After compensation with the optical density of the ACTB signal, the expression level in the vehicle control or 5 mM K^+^ is expressed as 1.0 (n = 4 for each). **: *p* < 0.01 vs. the vehicle control or 5 mM K^+^.

**Figure 10 ijms-23-08603-f010:**
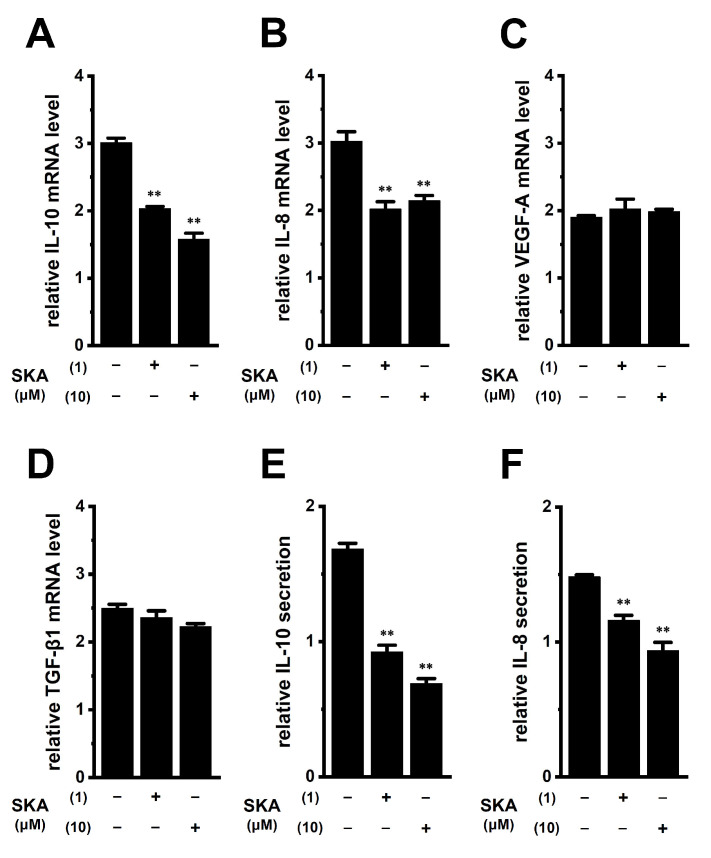
Effects of treatment with SKA-121 on expression levels of IL-10, IL-8, VEGF-A, and TGF-β1 transcripts and IL-10 and IL-8 secretion in human prostate cancer PC-3 cell-cultured medium-treated THP-1-derived M_2_ macrophages. (**A**–**D**): Real-time PCR examination of IL-10 (**A**), IL-8 (**B**), VEGF-A (**C**), and TGF-β1 (**D**) expression in PC-3 media-treated THP-1-derived M_2_ macrophages treated (+) or untreated (−) with 1 or 10 μM SKA-121 for 24 h. Relative mRNA expression in the group untreated with PC-3 medium is expressed as 1.0 (n = 4 for each). (**E**,**F**): Quantitative detection of IL-10 (**E**) and IL-8 (**F**) secretion by an ELISA assay in THP-1-derived M_2_ macrophages treated or untreated with SKA-121. Relative secretion in the group untreated with PC-3 medium is expressed as 1.0 (n = 4 for each). **: *p* < 0.01 vs. the vehicle control (−/−).

## Data Availability

Not applicable.

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
