# Peer review of "Downregulation of IL-8 and IL-10 by the Activation of Ca2+-Activated K+ Channel KCa3.1 in THP-1-Derived M2 Macrophages"

_ijms, 2022, doi:10.3390/ijms23158603_

Round 1

Reviewer 1 Report

I have no more comments.

Reviewer 2 Report

Within their revised manuscript the authors very well adressed all points raised within the first revision.

Additionally performed experiments and the changes within the main text significantly contribute to the understanding of the study and complement the work. The authors furthermore decided to remove parts of the data for a clearer presentation.  

Minor points:

The authors might check for the spelling within the figure description of Figure S1: Lane 2 à replace “alon” with “alone” and within figure description of Figure2 S2: Lane 4 à replace “24 h” with “24-h”. 

Overall impression: 

The work is technically sound and describes the presented data and methods in a clear and concise way. Hence, I recommend publishing the paper in its current form after correction of the spelling mistakes.

This manuscript is a resubmission of an earlier submission. The following is a list of the peer review reports and author responses from that submission.

Round 1

Reviewer 1 Report

The manuscript by Ohya et al focuses on the contribution of the calcium-activated potassium channel KCa3.1 in the expression and secretion of pro-and anti-inflammatory factors such as IL8 and IL10. Within their study, they unveiled a role of KCa3.1 in IL8 and IL10 expression and secretion, especially in response to environments possessing TME-like features such as high extracellular K+. Furthermore, the authors provide insights into the transcriptional regulation of IL8 and IL10.

Overall impression of the manuscript: The manuscript is well written in terms of style and language and might be of interest for cancer research, however some changes and additional points need to be addressed.

Since beeing based on a K+ channel, the study lacks proper characterization of the KCa3.1. Furthermore, the manuscript would significantly benefit from experiments focusing on K+ and Ca2+ ion levels, as well as electrophysiology to characterize the channel, since the authors very much focused on the transcriptional/genetic regulation. While they properly characterized and discussed possible pathways for regulation, the experiments performed using the spheroid culture medium seem incomplete. While the authors indeed discuss the outcome of their work for cancer therapy, they omit discussing the effect of their results on the cellular levels. Hence, the discussion rather resembles a repetition of the introduction and results than an interpretation. The paper would significantly benefit from a discussion that addresses possible effects of targeting KCa3.1 on the cellular level than on the global level. 

  • Line48/49: “KCa3.1 regulates the production of pro- and anti-inflammatory cytokines/chemokines and macrophage polarization.”

I encourage the authors to rephrase this sentence. KCa3.1 might contribute to the regulation, but might not be the main driving force.

  • The authors state that addition of KCa3.1 channel openers leads to elevation of intracellular Ca2+ and extracellular K+. It might be interesting to see if the results obtained from treatments with KCa3.1 openers might be reproducible upon elevation of cytosolic Ca2+ from intracellular Ca2+ stores or upon cell permeabilization.
  • The authors evaluated the presence of the KCa3.1 channel in M2 macrophages. However, they did not check for the channel functionality. Since the manuscript is based on the physiological activity of a K+ channel, it seems inevitable to perform electrophysiology to confirm channel functionality.
  • Figure 2B: The single cell trace of cell b within panel B shows increasing FURA signals from the beginning, which could be caused by pre-stimulation of the cells. Furthermore there is a high signal to noise ratio (especially in cell b). It would be good to show more individual cells.
  • Figure 2A&B: The authors did not state whether they used a perfusion system to add the channel opener or if the substances were injected by pipetting. However, since this (and other) experimental approaches are inevitable to reproduce experiments I highly encourage the authors to describe the methods more detailed.
  • By addition of the channel inhibitor TRAM-34 alone, the authors would be able to check for any basal activity of the channel before opening KCa3.1. Furthermore, upon measuring the maximum response of Ca2+, a quantitative analysis would be possible. This might be especially interesting in the future and would serve as a reference for testing the effect of other Ca2+-releasing pharmacological tools on KCa3.1 and/or IL8/IL10 expression.
  • In analogy to measuring Ca2+i, the study would significantly benefit from measurements of intra- or extracellular K+.
  • While the authors tested cell viability, they might also check for in- or decreased cell proliferation in response to channel opening/inhibition and high K+ex and lactate. Microscopy images of the cells in response to the mentioned stimuli would unveil any possible morphological changes.
  • Did the authors test higher concentrations of SKA-121 for the effect on the channel and why did they choose 1 µM?
  • Line 101: The authors mention simultaneous monitoring of DiBAC and FURA. However, they do not describe how they loaded the cells, if measured simultaneously. A description of the loading protocol (what was loaded first, which buffers were used), however, is inevitable to reproduce the measurements.
  • Line 120: The authors described the concentration of TRAM-34 to be 10 µM, but previously write that the concentration used was 1 µM.
  • Line 119: What do the authors mean by co-treatment? Did the authors apply the substances simultaneously i.e. from the beginning on? Please clarify within the methods section.
  • Supplementary Figure 1C/D: Within their controls, the authors did not check for the effect of 10 µM SKA-121 in response to normal (5mM K+ex) conditions. In summary, the controls should containing all conditions tested within Figure 4 and be presented the same way for easier read-out.
  • According to Figure 5D&E, channel opening fails to reduce IL8 and IL10 expression and secretion in response to high lactate levels. The authors might discuss any consequences for clinical applications and limitations of KCa3.1 channel openers based on high extracellular lactate or pH levels frequently found within the TME.
  • Line 152: Why did the authors use 0mM lactate as the control, while they stated that physiological levels should be around 1.5 – 3 mM?
  • If the addition of the channel opener in response to high K+, but not in response to high lactate reduces IL expression, the authors might should test a combination of these environmental factors.
  • I encourage the authors to revise the order of all panels in Figure 6 and 7, as they appear in different orders within the text.
  • The figures should be mentioned in a uniform way within the text (i.e Figure or Fig.).
  • The study would benefit from a more detailed discussion why SKA-121 failed to reduce IL expression in response to high lactate but not to high K+. Hence, possible mechanisms might be discussed.
  • Line 342: The authors mentioned Figure 5A,D, however it might be changed to Figure 3A,D, since Figure 5 does not show the respective data.
  • Line 365-386: “These results suggest that the down-regulation of IL-8 by the KCa3.1 activator in TAMs not only prevented cancer migration and metastasis but also reduced microvessel density in cancer”. The authors might rephrase this statement, as it represents a hypothesis, but was not measured.
  • Line 369: The figure reference 6A,D might be wrong.
  • Line 392-393: “Therefore, the PC-3-medium-induced up-regulation of IL-10 and IL-8 in TAMs may be attributed to microRNA(s) in cancer cells, promoting EMT in the TME.” In accordance with the author´s statement, it is necessary to provide mechanistic insights WHY the spheroid culture medium induces IL expression. Measuring the pH, Ca2+, K+ and lactate levels, hence, would be desirable. Furthermore, the authors hypothesize that miR-21 might contribute to IL expression. Measuring miR-21 levels within the spheroid culture medium would significantly underline the relevance of the experiments shown in Figure 10.
  • Line 406-410: “Severe acute respiratory syndrome coronavirus 2 (SARS-CoV-2) virus infection significantly increases the production of inflammatory cytokines and chemokines such as IL- 1β and IL-8. Macrophage KCa3.1 activation in coronavirus disease 2019 (COVID-19) may be responsible for the increased anti-inflammatory activity, which attenuates the severity of the SARS-CoV-2 cytokine storm.” This statement is too hypothetical. The authors should consider re-phrasing it as KCa3.1 might contribute, but is not likely to be the main regulating factor.
  • Line 506-509: “We demonstrated that KCa3.1 activators reversed TAM-mediated escape from anti-tumor immune surveillance (...) which prevented the metastatic potential and angiogenesis of cancer.“ The authors neither performed these experiments nor show these data, as they did not measure escape from the immune system or metastasis/angiogenesis. These events might represent a theoretical outcome of the results shown, but is not has not been measured. I would highly encourage the authors to omit such statements throughout the paper to prevent misinterpretation of the results.

Reviewer 2 Report

The results presented in the MS suggest that the KCa3.1 activator, SKA-121, may suppress production of  IL-8 and IL-10 in  THP-1-derived, M2-marophages    through  both ERK-CREB and JNK-c-Jun cascades, thereby reducing the pro-tumorigenic macrophage activity.

My comments and suggestions:

  1. The functional effects of only a single dose of SKA-121 (1μM) have been studied. It is important to show the dose-dependence (or independence) of SKA-121 effects on macrophages.
  2. The total (pro- tumorigenic or anti-tumorigenic) action of macrophages on the tumor is determined by the balance of the M2 and M1 functionalities opposing each other. Therefore, from a therapeutic point of view, it is important to evaluate (or, at least, to discuss) the action of SKA-121 not only on M2, but also on M1 macrophages generated from THP-1 cells under the influence of LPS and IFN-gamma.